# Treatment of HPV-Related Uterine Cervical Cancer with a Third-Generation Oncolytic Herpes Simplex Virus in Combination with an Immune Checkpoint Inhibitor

**DOI:** 10.3390/ijms24031988

**Published:** 2023-01-19

**Authors:** Masahiro Kagabu, Naoto Yoshino, Kazuyuki Murakami, Hideki Kawamura, Yutaka Sasaki, Yasushi Muraki, Tsukasa Baba

**Affiliations:** 1Department of Obstetrics and Gynecology, School of Medicine, Iwate Medical University, Shiwa 028-3695, Iwate, Japan; 2Division of Infectious Diseases and Immunology, Department of Microbiology, School of Medicine, Iwate Medical University, Shiwa 028-3694, Iwate, Japan

**Keywords:** cervical cancer, oncolytic virus, herpes simplex virus, oncolytic viral therapy, immune checkpoint inhibitor

## Abstract

Cervical cancer is one of the most common cancers in women. The development of new therapies with immune checkpoint inhibitors (ICIs) is being investigated for cervical cancer; however, their efficacy is not currently sufficient. Oncolytic virus therapy can increase tumor immunogenicity and enhance the antitumor effect of ICIs. In this report, the therapeutic potential of a triple-mutated oncolytic herpes virus (T-01) with an ICI for human papillomavirus (HPV)-related cervical cancer was evaluated using a bilateral syngeneic murine model. The efficacy of intratumoral (i.t.) administration with T-01 and subcutaneous (s.c.) administration of anti-programmed cell death ligand 1 (PD-L1) antibody (Ab) was equivalent to that of anti-PD-L1 Ab alone on the T-01-injected side. Moreover, combination therapy had no significant antitumor effect compared to monotherapy on the T-01-non-injected side. Combination therapy significantly increased the number of tumor specific T cells in the tumor. While T-01 could not be isolated from tumors receiving combination therapy, it could be isolated following T-01 monotherapy. Furthermore, T-01 had a cytotoxic effect on stimulated T cells. These results suggest that T-01 and anti-PD-L1 Ab partially counteract and therefore concomitant administration should be considered with caution.

## 1. Introduction

Cervical cancer is the fourth most common cancer in women, and the seventh most common of all human cancers. The global incidence of and mortality from cervical cancer in 2020 were approximately 604,000 and 341,000, respectively, according to the Agency for Research on Cancer (IARC) database [1]. Tumor metastasis is the most common cause of death in cervical cancer. Currently, radiotherapy and chemotherapy have been used to treat patients with advanced uterine cervical cancer [2], but with limited success. Moreover, the available chemotherapeutic agents have only limited efficacy against recurrent disease. Therefore, new therapeutic strategies for advanced and recurrent cervical cancers are urgently needed.

Cervical cancer is caused by persistent human papillomavirus (HPV) infection. HPV E6 and E7 induce a cascade of interleukin 6 (IL6) cytokine and T-cell signaling [3]. Increased IL6 subsequently induces myelo/monocyte infiltration, activates fibroblast inflammation, and disables antigen presentation. As a result, infiltration of regulatory T cells (Treg) and myeloid-derived suppressor cells (MDSC) occurs and programed cell death-ligand 1 (PD-L1) upregulation results in an immunosuppressed state. This immunosuppressed state enables the possibility of effective immunotherapy in cervical cancer [3], and some clinical trials are currently being conducted [4]. Clinical trials of immunotherapy for cervical cancer have been conducted to explore the application of immune checkpoint inhibitors (ICIs). The KEYNOTE-158 study has established a clinical benefit for pembrolizumab (humanized monoclonal immunoglobulin (Ig) G4 antibody directed against the programed cell death-1; PD-1 receptor) in cervical cancer with a 12.2% (95% confidence interval, 6.5% to 20.4%) overall response rate reported [5], results that are not fully satisfactory. The current treatment trend involves the establishment of ICI combination therapy rather than ICI monotherapy. Subsequent to KEYNOTE-158, the KEYNOTE-826 study was conducted as a phase 3 trial to evaluate the add-on effect of pembrolizumab to chemotherapy for unresectable or recurrent cervical cancer [6]. Pembrolizumab significantly prolonged progression-free survival (PFS) and overall survival compared with the placebo group; however, the results showed that pembrolizumab only prolonged PFS to 2.2 months. Virus-related cancers are immunotherapy targets for neoantigens derived from genetic mutations. However, ICI monotherapy did not show satisfactory efficacy against cervical cancer. Moreover, a major contributing factor to early resistance to ICI is the absent or low infiltration of tumor T cells, which is a characteristic of so-called “cold tumors”. Therefore, new therapies are needed that promote immune system priming with tumor neoantigens, induce effector T cell trafficking (“hot tumors”), and restore the immunosuppressive tumor microenvironment (TME) [7]. Thus, oncolytic virotherapy has been attracting increased attention as a promising method for enhancing ICI effects [8].

Oncolytic viral therapy is noted as a new strategy for the treatment of advanced cancers. Among therapeutic viruses, oncolytic herpes simplex virus (HSV) is the most remarkable. Oncolytic HSV is a genetically modified version of HSV where virus propagation is restricted to tumors, thereby killing infected tumor cells and activating antitumor immune responses [9]. Oncolytic HSV has been shown to shrink metastatic tumors by activating tumor immunity [10]. The ability to induce specific antitumor immunity enables the feasibility of combining oncolytic viruses with ICIs to treat cancer.

Previously, we reported the efficacy of the oncolytic HSV T-01 in human papillomavirus (HPV)-related cervical cancer models [11]. T-01 has a genomic structure similar to that of G47∆. In T-01, the α47 and γ34.5 loci are deleted, and the LacZ gene replaces the ICP6 gene [12,13]. G47∆ has shown efficacy in many malignant tumors [10,14,15,16,17,18]. Of these, G47∆ showed a strong antitumor effect on glioblastoma in clinical trials [19,20]. Based on these results, G47∆ was approved in Japan for glioblastoma. We have shown that T-01 has cytotoxic efficacy against cervical cancer cells, and that T-01 inhibits cervical cancer cell growth [11].

PD-1 has two types of ligand, PD-L1 and PD-L2 [21]. PD-1 is mainly expressed on T cells, while PD-L1 and PD-L2 are expressed on cancer and normal cells, respectively. Therefore, inhibition of PD-L1 with anti-PD-L1 antibody (Ab) does not cause T-cell activation in normal tissues. Anti-PD-L1 Ab has potentially fewer side effects than when both PD-1/PD-L1 and PD-1/PD-L2 are inhibited by anti-PD-1 Ab.

Here, we report the results of oncolytic HSV therapy with anti-PD-L1 Ab for cervical cancer. We investigated whether T-01 enhances the antitumor effect of an ICI in a bilateral tumor model.

## 2. Results

### 2.1. In Vivo Treatment of TC-1 Syngeneic Mice with T-01 and Anti-PD-L1 Ab

We investigated whether T-01 enhances the antitumor effect of ICIs in a bilateral tumor model (Figure 1). In this study the control group completed observations until day 25, with all mice in the control group reaching a humane endpoint by day 25. Among the treated groups, tumor growth on the T-01-injected side of the T-01, and PD-L1 Ab, and combination groups was significantly inhibited compared to the control group. However, tumor growth in the combination group was not significantly different from tumor growth in the PD-L1 Ab group (Figure 2a,b). On the non-injected side, the tumor growth was significantly reduced in the anti-PD-L1 and combination groups compared to the control group. Conversely, the tumor growth in the T-01 group was not significantly different than that in the control group. In addition, tumor growth was not significantly inhibited by combination therapy compared to PD-L1 Ab therapy (Figure 2a,c).

Six-week-old female C57BL/6 mice harboring bilateral subcutaneous TC-1 tumors of approximately 5 mm in diameter were treated with T-01, anti-PD-L1 antibody, or a combination thereof. The left-side tumor received an intratumoral dose of 1 × 10^5^ PFU of T-01 or a mock injection (PBS) in combination with subcutaneous injections with anti-PD-L1 antibody (100 μg on days 11 and 21).

### 2.2. Tumor-Specific and Oncolytic HSV-Specific CTLs Analysis

We analyzed tumor-infiltrating, tumor-specific and HSV-specific cytotoxic T lymphocytes (CTLs) 13 days after TC-1inoculation. In the injected side, the combination group showed significantly increased percentages of E7-specific CD8+ T-cells compared with those of the T-01 group (combination vs. T-01 = 0.96% vs. 0.48%, *p* < 0.001) and the PD-L1 Ab group (combination vs. PD-L1 Ab = 0.96% vs. 0.07%, *p* < 0.0001). In addition, the T-01 group showed significantly increased percentages of E7-specific CD8+ T-cells compared with those of the PD-L1 Ab group (T-01 vs. PD-L1 Ab = 0.48% vs. 0.07%, *p* < 0.01). In the non-injected side, the combination group showed significantly increased percentages of E7-specific CD8+ T-cells compared with those of the T-01 group (combination vs. T-01 = 0.54% vs. 0.17%, *p* < 0.05) and the PD-L1 Ab group (combination vs. PD-L1 Ab = 0.54% vs. 0.14%, *p* < 0.01). Conversely, there was no significant difference in the percentage of E7-specific CD8+ T-cells between the T-01 group and the PD-L1 Ab group (T-01 vs. PD-L1 Ab = 0.48% vs. 0.07%, not significant) (Figure 3a). Furthermore, the HSV-specific CTLs in tumor tissue were higher in the combination group than in the T-01 group. However, there was no significant difference in the percentage of HSV-specific CD8+ T-cells observed between the T-01 group and the combination group in the injected side (combination vs. T-01 = 0.36% vs. 0.28%) or the non-injected side (combination vs. T-01 = 0.23% vs. 0.16%) (Figure 3b). CTL of control group was not analyzed in this study. The present CTL analysis was designed to analyze whether the difference in treatment effect in each treatment group was due to an increase in CTL and not for comparison with the control group. Therefore, CTL analysis was not performed for the control group.

### 2.3. Quantification of Virus in Tumor Tissues

We quantified T-01 virus levels in tumor tissues of the T-01 and combination groups (Figure 4) at days 13 and 18. Day 13 is 2 days after the second virus administration and 2 days after the anti-PD-L1 antibody administration. Day 18 is 2 days after the third virus administration and 7 days after the an-ti-PD-L1 antibody administration. On the injected side, the titer of T-01 was significantly higher in the T-01 group than in the combination group at day 13 (T-01 vs. combination = 35.8 PFU/g vs. below detection limit, *p* < 0.05). Comparatively, the titer of T-01 on the non-injected side was below the detection limit in both the T-01 and combination groups at day 13 (Figure 4). Moreover, the titer of T-01 on the injected side decreased on day 18 compared to day 13. There was no significant difference between the combination and T-01 groups at day 18 (T-01 vs. combination = 9.6 vs. 3.0 PFU/g, not significant) (Figure 4).

### 2.4. Measurement of T-01 Cytotoxicity of Phytohemagglutinin (PHA)-Stimulated Lymphocytes

We examined the cytotoxic effect of T-01 on activated lymphocytes. Although T-01 was not cytotoxic to unstimulated lymphocytes, the cell survival rate was significantly decreased in PHA-stimulated lymphocytes compared with unstimulated lymphocytes (Figure 5).

## 3. Discussion

Oncolytic viral therapy is a highly promising new therapeutic strategy for cancer treatment. It is currently approved by the United States Food and Drug Administration as a treatment for malignant melanoma and other cancers [22]. Moreover, oncolytic viral therapy with ICIs has been investigated in numerous clinical trials [6]. Nevertheless, only two reports of oncolytic viral monotherapy for cervical cancer have been published to date [11,23]. This is the first report on the efficacy of oncolytic HSV therapy in combination with an ICI for cervical cancer using a bilateral tumor model. In antitumor immunotherapy, the bilateral model is extremely useful for assessing the effect of local antitumor treatments. While there are multiple reasons for this, the ability to examine two tumors in the same animal enables direct evaluations of the advantages and drawbacks of combination therapies.

We observed herein that T-01 with anti-PD-L1 Ab therapy significantly reduced tumor growth compared to control in the injected side. We have previously reported that tumor-specific CTLs are increased by T-01 therapy [11]. In this study, we demonstrated an increase in tumor-infiltrating CTLs in the T-01 monotherapy and anti-PD-1 Ab combination therapy groups (Figure 3a). However, the antitumor effect was not strong enough to be deemed a synergistic effect despite the combined use of an ICI. Combination therapy with PD-L1 Ab drugs using vaccinia virus has been reported to increase tumor-infiltrating CTLs and to have an antitumor effect [24]. Notably, G47Δ has two antitumor mechanisms: one is through intratumoral proliferation and cytotoxicity, and the other is reducing tumor size through antitumor immunity activated by the destruction of tumor cells [25]. In addition, G47Δ induces the expression of MHC class I on the surface of infected cells, which enhances the immune response. In this study, T-01 / ICI combination therapy showed a decrease in the quantity of virus in the tumor compared to virus monotherapy (Figure 4), while HSV-specific CTL infiltration was confirmed (Figure 3b). This suggests that the anti-tumor effect may have been attenuated as a result of reduced tumors directly destroyed by viral infection.

This is the first report of a bilateral model of cervical cancer in mice using TC-1 cells. G47Δ has been reported to have a significant antitumor effect on the untreated side compared to the control group in bilateral models [26]. In our study, tumor-specific tumor-infiltrating lymphocytes were increased in the combination group compared with the T-01 group and the PD-L1 Ab group (Figure 3a). G47Δ is effective against a variety of tumors, including certain hematological tumors [27]. In this study, we examined the cytotoxicity of T-01 toward lymphocytes and found T-01 produced a cytotoxic effect on stimulated lymphocytes (Figure 5), which acquire a proliferative capacity. In this experiment, the cytotoxic effect of T-01 on proliferating T cells is demonstrated. This study did not analyze proliferative and non-proliferative CTLs within the tumor. The results of the PHA experiments in the present study support the hypothesis that the effects of T-01 on lymphocytes differ according to their proliferative potential. 

The advent of ICIs in cancer treatment has further advanced the study of cancer immunity. Previously, the effect of checkpoint inhibitors was commonly thought of as “driver and brake”. However, the phenomenon of T cell “exhaustion” has been attracting recent attention. T cells can become dysfunctional (exhausted) through repeated stimulation in cancer tissue or in in vitro studies. This is considered to be one of the factors that may diminish the therapeutic effect of cancer immunotherapy [28]. In our study we speculated that T cells may have become exhausted following concomitant stimulation with T-01 and the associated tumor antigens, which may attenuate the effect of checkpoint inhibitors. Nevertheless, it has been reported that combination therapy with G47Δ and the ICI anti-cytotoxic T-lymphocyte-associated protein 4 (CTLA-4) antibody showed a stronger antitumor effect than the individual agents [27,29]. CTLA-4 and PD-1 have distinct mechanisms of action: CTLA-4 suppresses T-cell activation during the priming phase; in contrast, PD-1 suppresses T-cell activity in peripheral tissues primarily through an intrinsic cellular mechanism [26,27]. Whether or not G47Δ infects and causes cytotoxicity in activated T cells during the priming phase needs further validation.

The present study used a mouse model to validate a combination therapy with an ICI and G47Δ. One of the limitations of the present study is its focus on tumor-infiltrating lymphocytes only. Thus, it is necessary to study other immune mechanisms in the future. Notably, it has been reported that the immune mechanisms in mice differ from those of humans. The results of this study do not necessarily ensure the efficacy of the therapy in humans. However, clinical trials involving a combination of Talimogene laherparepvec (T-VEC), a double-mutated, second-generation oncolytic HSV-1, and ICIs showed an additive effect compared to the individual agents [29]. For virus-associated cervical cancer options for chemotherapy and ICI combination therapy are limited [6]. Currently, there is sufficient scientific basis to support a clinical trial examining the effects of combining G47Δ with ICI therapy.

## 4. Materials and Methods

### 4.1. Ethics Statement

All animal experiments were performed in accordance with the recommendations in the *Guidelines for Proper Conduct of Animal Experiments* established by the Science Council of Japan and the guidelines established by the Committee on the Ethics of Animal Experiments (CEAE) at Iwate Medical University. The Iwate Medical University CEAE gave approval for all procedures involving mice (permit number 27-004). The mice were euthanized as a humane endpoint when any of the following conditions were observed: tumor volumes exceeding a maximum of 1500 mm^3^, upon signs of health deterioration, or weight loss exceeding 20%.

### 4.2. Cell Lines and Virus

TC-1 cells were obtained by cotransformation of primary C57BL/6 mouse lung epithelial cells with HPV16 E6 and E7 and an activated ras oncogene, as described previously [30]. The TC-1 cells were a gift from Dr. T. C. Wu (Johns Hopkins University, Baltimore, MD, USA). TC-1 cells were cultured in RPMI 1640 (Life Technologies Co., Carlsbad, CA, USA) supplemented with 2 mM L-glutamine, 10 mM 4–(2–hydroxyethyl)–1–piperazineethanesulfonic acid (Dojindo Molecular Technologies, Inc., Kumamoto, Japan), 200 U/mL penicillin, 0.2 mg/mL streptomycin, and 10% fetal bovine serum (FBS). The cells were cultured under standard conditions (37 °C, 5% CO_2_ atmosphere).

The details of T-01 (kindly provided by T. Todo, University of Tokyo, Japan) construction have been published previously [10,11]. Viral stocks were prepared by releasing the virus from infected Vero cells using a freeze–thaw/sonication regimen, followed by ultracentrifugation [30].

### 4.3. In Vivo Treatment of TC-1 Tumors with T-01 and Anti-PD-L1 Antibody (Ab)

Five-week-old female C57BL/6 mice (15–18 g) (CLEA Japan Inc., Tokyo, Japan) were used in protocols approved by the Committee on the Ethics of Animal Experiments (CEAE) at Iwate Medical University. In the TC-1 model, 1 × 10^5^ cells in 0.1 mL PBS were subcutaneously injected bilaterally in the flank of C57BL/6 mice. Animals received an intratumoral dose of 1 × 10^5^ PFU of T-01 suspended in 50 µL of PBS 7, 11 and 16 days after TC-1 injection. Rat anti-mouse PD-L1 monoclonal Ab (subclass: IgG2b, clone: 10F.9G2, Bio Legend Inc., San Diego, CA, USA) was used as an immune checkpoint inhibitor. Anti-PD-L1 Ab was injected subcutaneously between the left and right tumors 11 and 21 days after TC-1 injection. The antitumor effect in the mouse model of cervical cancer was investigated in the following four groups: control group, T-01 group, PD-L1 Ab group, and combination group. The T-01 and combination groups were administered T-01 to the tumor on the left side. The PD-L1 Ab group and combination group were administered anti-PD-L1 Ab subcutaneously between the left and right tumors. The control group was administered PBS to the tumor on the left side. T-01 or PBS were administered intratumorally at 4- to 5-day intervals starting on day 7 after TC-1 implantation. Anti-PD-L1 Ab was administered subcutaneously between the dorsal tumors 11 and 21 days after TC-1 inoculation. The anti-PD-L1 Ab was administered at a dose of 100 μg/100 μL per inoculation (Figure 1). Tumor volumes were calculated using the formula V = ½ L × W^2^, where L is the length (longest dimension) and W is the width (shortest dimension) [30].

### 4.4. Tissue Collection and Cell Isolation

Tumor tissues were collected from the mice in each treatment group. These tumor tissues were crushed with a gentleMACS Dissociator (Miltenyi Biotec Inc., Bergisch Gladbach, Germany) using a Tumor Dissociation Kit (Miltenyi Biotec Inc.), and cells were subsequently isolated. The number of viable cells was measured using a TC20TM fully automated cell counter (Bio Rad Laboratories, Hercules, CA, USA).

### 4.5. Tumor-Specific and Oncolytic HSV-Specific CTLs Analysis

Since TC-1 cells express the HPV E7 tumor antigen [31], cytotoxic T cells (CTLs) against TC-1 were analyzed by counting E7-specific CTLs, and CTLs against T-01 were measured by a tetramer assay using glycoprotein B (gB), one of the HSV envelope proteins, 13 days after T-01 inoculation. Cells isolated from tumor tissues were suspended in staining buffer (RPMI 1640 with 200 U/mL penicillin, 0.2 mg/mL streptomycin, and 3% FBS) at a concentration of 1 × 10^6^ cells/100 μL, and stained with phycoerythrin (PE)-labeled HPV16 E7 tetramer (MBL International Corporation, Nagoya, Japan) or PE-labeled HSV gB tetramer (MBL International Corporation), allophycocyanin (APC) labeled anti-CD3 antibody (clone: 17A2, Bio Legend Inc.), fluorescein isothiocyanate (FITC) labeled anti-CD8 antibody (clone: 53–6.7, Bio Legend), and peridinin-chlorophyll protein complex with cyanin-5.5 (PerCP-Cy5.5)-conjugated anti-CD45 antibody (clone: I3/2.3, Bio Legend Inc.) for 30 min in the dark. The cells were washed with staining buffer, measured with a FACSCalibur (BD Biosciences, Franklin Lakes, NJ, USA), and analyzed with Cell Quest Pro software version 6.0 (BD Biosciences). All flow cytometer measurements were performed under the same conditions.

### 4.6. Quantification of Virus in Tumor Tissue

Tumor tissues (on 13 and 18 days after T-01 inoculation) were excised from mice and crushed using gentleMACS using a Tumor Dissociation Kit (Miltenyi Biotec Inc.), followed by centrifugation at 800× *g* for 5 min at 4 °C. The supernatant was collected as a virus sample and stored at −80 °C until use.

The quantification (plaque titration) of T-01 virus contained in the tumor tissue was performed as described previously [32]. Briefly, Vero cells were maintained in Dulbecco’s Modified Eagle Medium (DMEM; Life Technologies Co., Ltd. Tokyo, Japan) supplemented with 10% FBS, 20 units/mL penicillin, and 0.2 mg/mL streptomycin at 37 °C under 5% CO_2_; they were seeded in a 6-well flat culture plate (ThermoFisher Scientific Inc., Waltham MA, USA) (1.5 × 10^6^ cells/well) and incubated for 24 h. The virus samples were serially diluted (10-fold; up to 1 × 10^−3^) with Minimum Essential Medium (MEM; Life Technologies Co., Ltd.) and infected to the cells (250 μL/well). The infected cells were incubated in DMEM (Nissui Pharmaceutical Co., Ltd., Tokyo, Japan) containing 1% Sea plaque agarose (Lonza Rockland, Inc., Rockland, ME, USA) and 2% FBS. At 5 days post-infection, the cells were stained with crystal violet (Sigma-Aldrich, St. Louis, MO, USA) and the number of plaques were counted. The titer (plaque forming unit, PFU) was determined as the value per gram of tumor tissue.

### 4.7. Measurement of T-01 Cytotoxicity in Phytohemagglutinin (PHA) Lymphocytes

After harvesting the spleen from a naïve mouse it was passed through a 70 µm-cell strainer (BD Biosciences). Lymphocytes were obtained by lysing erythrocytes in ACK buffer (0.826% NH_4_Cl, 0.1% KHCO_3_, 0.0037% EDTA_2_Na, pH 7.3). The lymphocytes were then seeded into 96-well culture plates at 1 × 10^6^ cells/well and cultured in the presence or absence of 1% PHA-M (Thermo Fisher Scientific Inc. Tokyo, Japan) for 24 h. The cells were infected with T-01 at 1 × 10^6^ PFU and cell viability was measured 24 h after infection. Cell viability was evaluated using a chromogenic kit (CellTiter Aqueous 96; Promega, Madison, WI, USA), and colorimetric detection was carried out in a microplate reader (Tecan Austria GmbH, Groedig, Austria). Cytopathic effects were documented using an optical microscope (Nikon, ECLIPSE, TE300, Tokyo, Japan). Cellular viability was measured by the method described above.

### 4.8. Statistical Analysis

All data are expressed as means ± standard deviation (SD). The statistical analyses for data were performed using one-way analysis of variance (ANOVA). If the ANOVA resulted in a significant difference, Tukey’s multiple comparison test was performed. A difference with *p* < 0.05 was considered significant. The Prism 8.0 software program (GraphPad Software Inc., La Jolla, CA, USA) was used for all statistical analyses.

## 5. Conclusions

To the best of our knowledge, this is the first report of triple-mutated, third-generation oncolytic HSV therapy in combination with an ICI for cervical cancer using a TC-1 syngeneic bilateral model. It was shown that oncolytic HSV with an ICI has antitumor efficacy against a cervical cancer model, and that oncolytic HSV inhibited cervical cancer cell growth. G47Δ was approved for the treatment of glioblastoma in Japan in 2021. These results demonstrate that oncolytic HSV and anti-PD-L1 Ab partially counteract and therefore concomitant administration should be considered with caution.

## Figures and Tables

**Figure 1 ijms-24-01988-f001:**
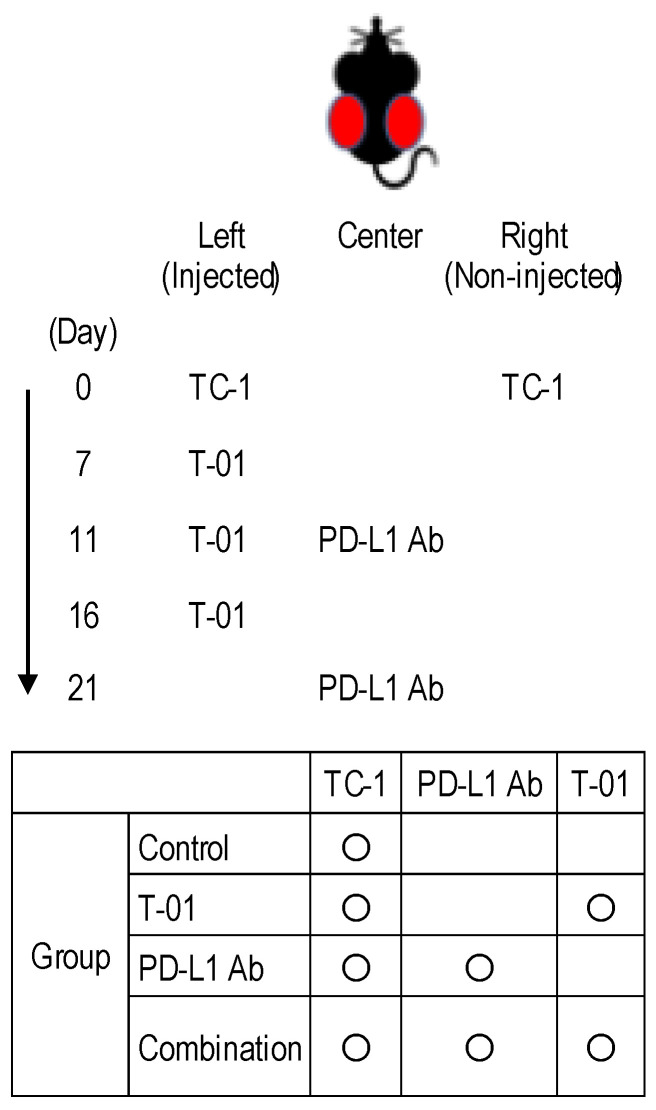
Experimental design of T-01 treatment in combination with PD-L1 inhibition in a murine bilateral subcutaneous TC-1 tumor mode.

**Figure 2 ijms-24-01988-f002:**
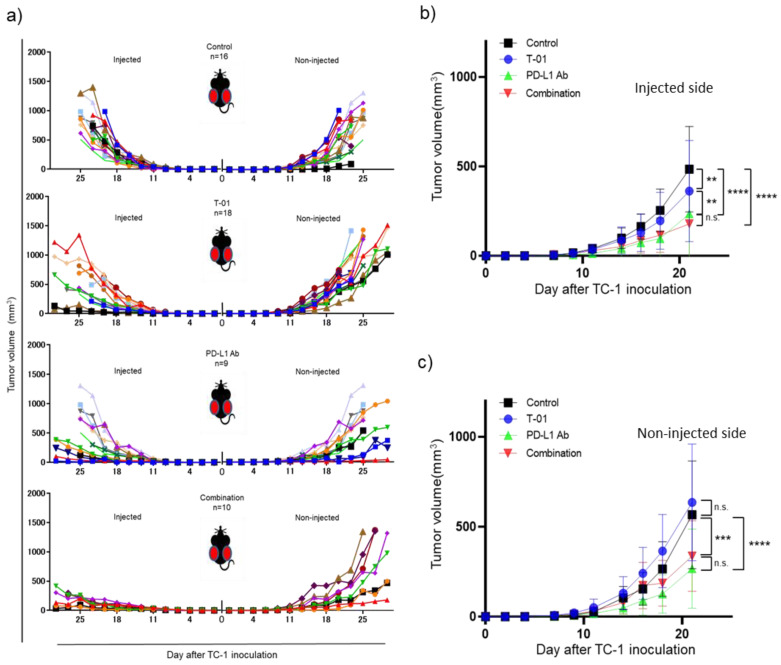
Efficacy of T-01 in combination with PD-L1 inhibition in a murine bilateral subcutaneous TC-1 tumor model. (**a**): Individual tumor growth curves of T-01 tumors. (**b**): Growth of TC-1 tumors in the T-01-injected side in animals. The results are presented as the mean ± SD (*n* = 9–18 per group) (** *p* < 0.01, *** *p* < 0.001, **** *p* < 0.0001 n.s.: not significant). (**c**): Growth of TC-1 tumors in the non-injected side in animals. These results are presented as the mean ± SD (*n* = 9–18 per group) (** *p* < 0.01, *** *p* < 0.001, **** *p* < 0.0001 n.s.: not significant).

**Figure 3 ijms-24-01988-f003:**
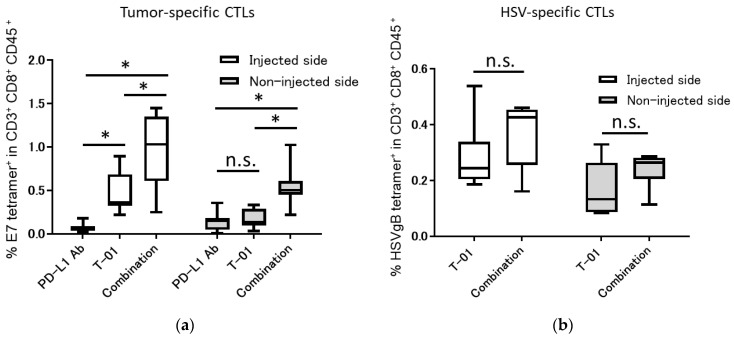
Tumor-specific and HSV-specific CTLs in bilateral subcutaneous TC-1 tumors treated with the combination of T-01 and anti-PD-L1 Ab. Thirteen days after tumor implantation, tumors were harvested and characterized for the presence of E7-specific CD8+ T-cells and HSV- specific CD8+ T-cells using E7-specific tetramer and HSV gB tetramer staining followed by flow cytometry analysis. (**a**): Tumor-infiltrating E7-specific CD8+ T-cells were analyzed by flow cytometry. (**b**): Tumor-infiltrating HSV-specific CD8+ T-cells were analyzed by flow cytometry. The results are presented as boxplots, which display the dataset based on the five-number summary: minimum, maximum, sample median, and the first and third quartiles (*n* = 8 per group). A two-way ANOVA followed by Sidak’s multiple comparisons test was used to determine statistical significance (* *p* < 0.05, n.s.: not significant).

**Figure 4 ijms-24-01988-f004:**
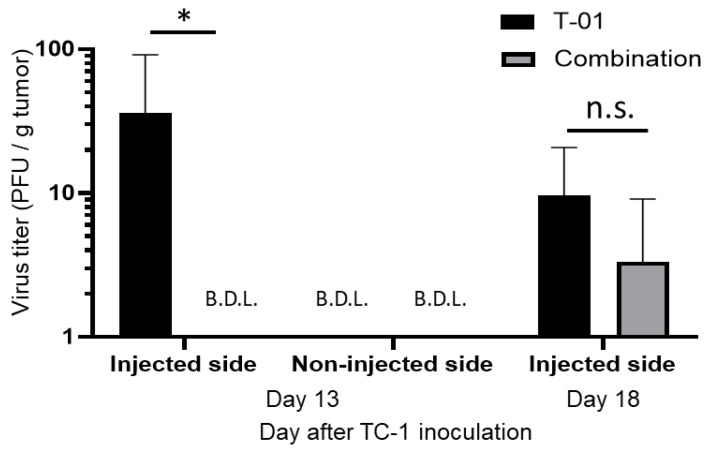
Analysis of viral titers in TC-1 tumor. Thirteen and eighteen days after tumor implantation, quantification of T-01 in the tumors of the T-01 and combination groups was performed. Day 13 is 2 days after the second virus administration and 2 days after the anti-PD-L1 antibody administration. Day 18 is 2 days after the third virus administration and 7 days after the anti-PD-L1 antibody administration. These results are presented as the mean + SD (*n* = 3 per group). The Mann–Whitney test was used to determine statistical significance (* *p* < 0.05, n.s.: not significant).

**Figure 5 ijms-24-01988-f005:**
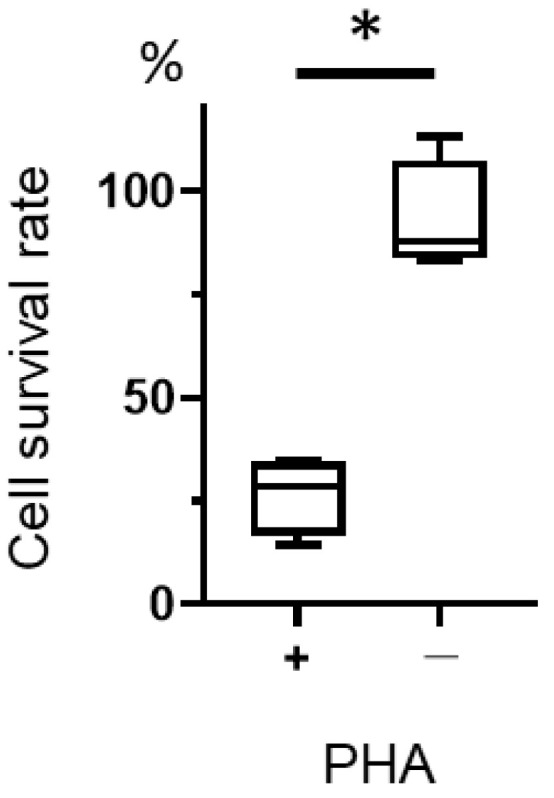
Cytotoxicity of T-01 against activated lymphocytes. Lymphocytes were isolated from the spleens of naïve mice, cultured in the presence or absence of PHA, and then infected with T-01. Cell survival was assayed 24 h after infection. The results are presented as boxplots, which display the dataset based on the five-number summary: minimum, maximum, sample median, and the first and third quartiles. The Student’s *t*-test was used to determine statistical significance (* *p* < 0.05). The experiments were performed four times. These data are the mean of 4 experiments.

## Data Availability

All relevant data are included in the manuscript or can be obtained from the authors on reasonable request.

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
