# Peer review of "Treatment of HPV-Related Uterine Cervical Cancer with a Third-Generation Oncolytic Herpes Simplex Virus in Combination with an Immune Checkpoint Inhibitor"

_ijms, 2023, doi:10.3390/ijms24031988_

Round 1
Reviewer 1 Report
Kagabu and colleagues investigated the efficacy of combined treatment with an oncolytic herpes simplex virus (oHSV) and anti-PD-L1 antibody in a murine model for cervical cancer. The presented results are interesting. In particular, the observation that oHSV killed cytotoxic T lymphocytes, thereby diminishing the effect of ICI treatment, is remarkable and relevant for future development of immunotherapy of cervical cancer.
The authors should address the following issues:
General:
The main conclusion presented in section 5 that “These results demonstrate that oncolytic HSV with ICIs should be considered for further development as a therapeutic strategy for cervical cancer” (lines 364-366) is incorrect. This conclusion is based on observations made by others using different oHSVs than were used in the present study. In contrast, the results presented in this manuscript call for caution when combining oHSV with ICIs.
Also the conclusion presented in the abstract (lines 26-28) that “These results suggest that improved combination therapy can be realized by optimizing the timing of administration […]” is not supported by the data presented in the manuscript. While it is the hope of the authors that the observed counterproductive effect of adding oHSV to anti-PD-L1 treatment can be overcome by changing the treatment schedule, this is not investigated. There is thus no support for this suggestion. All that can be concluded from the work is that concomitant administration is probably best avoided.
Specific:
Tumor growth inhibition experiment (Figures 1 and 2):
Why is the efficacy of oHSV monotherapy not tested against the control treatment? All other comparisons are made, but this comparison is not done. The figure suggests that oHSV monotherapy was effective.
Figure 2a shows that several animals were sacrificed before day 25, probably because their tumors grew too large. Figures 2b and 2c present mean tumor volumes until day 25. This is not correct; means cannot be calculated anymore when the fastest growing tumors are removed from the experiment. Please redraw these figures, with lines not extending beyond the day at which the first animal in a treatment group was sacrificed.
Please check if the values shown in these figures are accurate. E.g., figure 2a shows that at day 25 the largest tumor in the anti-PD-L1 group was approx. 1300mm3. This is difficult to match with the mean tumor volume of approx. 150mm3 with invisibly small SD at this point in figure 2b.
Please explain why a large variation in group sizes, ranging from 9 to 18 animals, was included in the experimental design; and indicate in the figure legend what were the group sizes for the individual treatment groups.
Please also explain why in the control group, PBS was injected in the right side tumor, while T-01 in the oHSV and combination treatments was injected in the left side tumor (lines 292-295). Is there a reason for this design; and could this have an effect on the outcome?
The sentence on lines 103-105 describes the opposite of what is shown in the figure. Probably, the sentence should read that tumor growth was reduced in the anti-PD-L1 and combination groups compared to the control group.
The statement on lines 106-107 that “tumor growth was significantly inhibited by PD-L1 Ab therapy compared to combination therapy” is wrong. The data show that T-01 injection inhibited the effect of the PD-L1 Ab.
CTL analysis (Figure 3):
In the presented experiment, the PBS control groups are missing. These should be included to judge the effect of the treatments, in particular the mono-treatments. Also, the HSV-specific CTL data from the anti-PD-L1 group are not shown. One might argue that there cannot be any HSV-specific CTL in non-virus injected tumors, but it is better to show this as it is the proper negative control. Adding these data should not be any problem for the authors, because it is clear that they have the samples for this analysis available.
There seems to be a discrepancy in the numbering of the days in the experiment. CTL were isolated from tumors “Eighteen days after tumor implantation” (line 147) and “13 days after T-01 inoculation” (line 128). This suggests that the virus was injected 5 days after tumor implantation. However, according to the experimental scheme presented in Figure 1 this was first given on day 7. In addition, since virus injection was repeated (days 7, 11, 16) the CTL analysis was probably 2 days after the last virus injection. For anti-PD-L1 treatment the schedule was different. Thus, CTL were analyzed in the compared treatment groups at different times after administration. Please provide a more clear description of the exposure times to the experimental agents before analysis; and discuss if different exposure times could influence the results.
Virus titer experiment (Figure 4):
The way this experiment is presented in figure 4 is unnecessarily complicated. It is not immediately clear that panel A is for day-13 only and panel B is for injected tumors only. The same data (injected tumors on day 13) are now duplicated in both panels. To allow direct interpretation, it would be better to present the data in a single panel and include the day-18 results of the non-injected tumors.
For this experiment, virus was released using the gentleMACS (lines 324-326). Please explain why this method was used; give details about the kit used; and provide support that the method efficiently isolates oHSV. The gentleMACS is commonly used for gentle dissociation of cells from tissues, to isolate intact cells for FACS analysis or culture. This casts some doubts if it will efficiently release oHSV from the cells into the cleared supernatant that was used for virus titration.
Further, please provide the detection limit of the assay used. Also, are the day-18 PFU values significantly above the detection limit? This influences interpretation of the findings.
By the way, time-points 13 and 18 are given relative to the tumor transplantation day. For interpretation, it is more relevant to present this relative to the day of virus injection.
Discussion section:
From the observation that combination treatment reduced the amount of virus in tumors, while virus-specific CTL were detected, it is concluded on lines 214-216: “This suggests that the immune activation toward HSV and the tumor attenuated the antitumor effect of T-01 monotherapy.” The observation does not suggest this at all. In contrast, the experiment shown in Figure 2 showed that combination treatment was more effective than oHSV alone, on injected and non-injected tumors. By the way, the effect of PD-L1 Ab on HSV-specific CTL was not significant anyway (Figure 3b).
The speculation on lines 236-239 about a presumed exhaustion of T cells upon “concomitant stimulation with T-01 and the associated tumor antigens” does not seem to be supported by the presented data. The experiment shown in Figure 5 was done in the absence of tumor cells. TAAs do thus not appear to play a role in the reduced recovery of PHA-stimulated lymphocytes upon T-01 infection.
Minor:
The description of the effects of E6/E7 on the immune system (lines 42-47) require supporting citations.
The statement that “Virus-related cancers are immunotherapy targets for neoantigens derived from cancer testis antigens […]” needs correction and explanation. (1) The neoantigens are the targets; (2) while this is evident for viral antigens, it is not immediately clear why virus-induced transformation is associated with cancer testis antigen expression.
On line 67, the authors probably don’t mean to say that therapies are needed that “restore” the immunosuppressive TME. Presumably, they mean “overcome”, “counteract”, OSLT.
Citation #10 does not sufficiently support the statement on lines 74-75 that oHSV shrank metastatic tumors by activating tumor immunity. In that study, oHSV armed with immune stimulatory transgenes were used. Best results were obtained when multiple immune stimulatory transgenes were combined. Effects of control empty oHSV on contralateral tumors were not significant.
On lines 76-77, the suggestion is made that oHSV can turn cold tumors into hot tumors. The citation #11 to support this is however work with another type virus, not HSV. Please provide a relevant citation or delete the statement.
On line 176, “T0-1” should be “T-01” and “of” should be “on”.
In the legend of Figure 5, please clarify if “The experiments were performed four times” (line 188) means the data are the mean of 4 experiments; or are from one typical experiment of four done.
In the discussion section 3, all references to figures are wrong. 2 should be 3; 3 should be 4; and 4 should be 5.
The “2” in “W2” on line 299 should be written in superscript.
Reference #32 on line 300 is superfluous; and probably not the appropriate citation for the applied formula, which was common practice long before the cited publication.
Lines 346-347: repeated sentence.
Line 349: Tecon should read Tecan.
Author Response
Professor Maurizio Battino,
Editor-in-Chief, International Journal of Molecular Sciences
Dear Dr. Battino and reviewers
Thank you for inviting us to submit a revised draft of our manuscript entitled, “Treatment of HPV-related uterine cervical cancer with a third-generation oncolytic herpes simplex virus in combination with an immune checkpoint inhibitor.” to International Journal of Molecular Sciences. We also appreciate the time and effort you and each of the reviewers have dedicated to providing insightful feedback on ways to strengthen our paper. Thus, it is with great pleasure that we resubmit our article for further consideration. We have incorporated changes that reflect the detailed suggestions you have graciously provided. We also hope that our edits and the responses we provide below satisfactorily address all the issues and concerns you and the reviewers have noted.
To facilitate your review of our revisions, the following is a point-by-point response to the questions and comments delivered in your letter.
Reviewer 1:
Kagabu and colleagues investigated the efficacy of combined treatment with an oncolytic herpes simplex virus (oHSV) and anti-PD-L1 antibody in a murine model for cervical cancer. The presented results are interesting. In particular, the observation that oHSV killed cytotoxic T lymphocytes, thereby diminishing the effect of ICI treatment, is remarkable and relevant for future development of immunotherapy of cervical cancer.
The authors should address the following issues:
General:
The main conclusion presented in section 5 that “These results demonstrate that oncolytic HSV with ICIs should be considered for further development as a therapeutic strategy for cervical cancer” (lines 364-366) is incorrect. This conclusion is based on observations made by others using different oHSVs than were used in the present study. In contrast, the results presented in this manuscript call for caution when combining oHSV with ICIs.
Also the conclusion presented in the abstract (lines 26-28) that “These results suggest that improved combination therapy can be realized by optimizing the timing of administration […]” is not supported by the data presented in the manuscript. While it is the hope of the authors that the observed counterproductive effect of adding oHSV to anti-PD-L1 treatment can be overcome by changing the treatment schedule, this is not investigated. There is thus no support for this suggestion. All that can be concluded from the work is that concomitant administration is probably best avoided.
Specific:
Tumor growth inhibition experiment (Figures 1 and 2):
Why is the efficacy of oHSV monotherapy not tested against the control treatment? All other comparisons are made, but this comparison is not done. The figure suggests that oHSV monotherapy was effective.
Thank you for your comments.
oHSV is T-01. The efficacy of oHSV(T-01) monotherapy
The administration and effects of T-01 are presented in Figures 1 and 2.
Figure 2a shows that several animals were sacrificed before day 25, probably because their tumors grew too large. Figures 2b and 2c present mean tumor volumes until day 25. This is not correct; means cannot be calculated anymore when the fastest growing tumors are removed from the experiment. Please redraw these figures, with lines not extending beyond the day at which the first animal in a treatment group was sacrificed.
Thank you for your comment. I have corrected Figures 2b and 2c.
Please check if the values shown in these figures are accurate. E.g., figure 2a shows that at day 25 the largest tumor in the anti-PD-L1 group was approx. 1300mm3. This is difficult to match with the mean tumor volume of approx. 150mm3 with invisibly small SD at this point in figure 2b.
Thank you for your comment. I have corrected Figures 2b and 2c.
Please explain why a large variation in group sizes, ranging from 9 to 18 animals, was included in the experimental design; and indicate in the figure legend what were the group sizes for the individual treatment groups.
Thank you for your comment. Sample sizes are already listed in Fig. 2a. This experiment was started with 4~6 animals per group and was conducted three times. Tumor development was checked before starting treatment, and tumor-free animals were excluded.
Please also explain why in the control group, PBS was injected in the right side tumor, while T-01 in the oHSV and combination treatments was injected in the left side tumor (lines 292-295). Is there a reason for this design; and could this have an effect on the outcome?
Thank you for your comment. PBS was injected the left tumor. I have corrected the misstatement.
The sentence on lines 103-105 describes the opposite of what is shown in the figure. Probably, the sentence should read that tumor growth was reduced in the anti-PD-L1 and combination groups compared to the control group.
Thank you for your comment. I have corrected the misstatement.
The statement on lines 106-107 that “tumor growth was significantly inhibited by PD-L1 Ab therapy compared to combination therapy” is wrong. The data show that T-01 injection inhibited the effect of the PD-L1 Ab.
Thank you for your comment. I have corrected Figures 2c and the sentences.
CTL analysis (Figure 3):
In the presented experiment, the PBS control groups are missing. These should be included to judge the effect of the treatments, in particular the mono-treatments. Also, the HSV-specific CTL data from the anti-PD-L1 group are not shown. One might argue that there cannot be any HSV-specific CTL in non-virus injected tumors, but it is better to show this as it is the proper negative control. Adding these data should not be any problem for the authors, because it is clear that they have the samples for this analysis available.
Thank you for your comment. We did not perform a CTL analysis of PBS control groups. Cell isolation with gntleMACS and CTL analysis with FACS are time critical. Therefore, it is not possible to process a large number of samples at the same time. We do not have the equipment and staff to make this possible. We did not perform the HSV-specific CTL analysis from the anti-PD-L1 group. We do not have the HSV-specific CTL data from the anti-PD-L1 group.
There seems to be a discrepancy in the numbering of the days in the experiment. CTL were isolated from tumors “Eighteen days after tumor implantation” (line 147) and “13 days after T-01 inoculation” (line 128). This suggests that the virus was injected 5 days after tumor implantation. However, according to the experimental scheme presented in Figure 1 this was first given on day 7. In addition, since virus injection was repeated (days 7, 11, 16) the CTL analysis was probably 2 days after the last virus injection. For anti-PD-L1 treatment the schedule was different. Thus, CTL were analyzed in the compared treatment groups at different times after administration. Please provide a more clear description of the exposure times to the experimental agents before analysis; and discuss if different exposure times could influence the results.
Thank you for your comment. The data presented manuscript are based on the results using tumors of 13 days after T-01 inoculation. CTL data of day 18 tumors is not posted. I have corrected the misstatement.
Virus titer experiment (Figure 4):
The way this experiment is presented in figure 4 is unnecessarily complicated. It is not immediately clear that panel A is for day-13 only and panel B is for injected tumors only. The same data (injected tumors on day 13) are now duplicated in both panels. To allow direct interpretation, it would be better to present the data in a single panel and include the day-18 results of the non-injected tumors.
Thank you for your comment. I have corrected Figures 4 and the sentences. However, we have not analyzed the non-injected tumors of D18. Because we did not detect the virus on D13.
For this experiment, virus was released using the gentleMACS (lines 324-326). Please explain why this method was used; give details about the kit used; and provide support that the method efficiently isolates oHSV. The gentleMACS is commonly used for gentle dissociation of cells from tissues, to isolate intact cells for FACS analysis or culture. This casts some doubts if it will efficiently release oHSV from the cells into the cleared supernatant that was used for virus titration.
Thank you for your comments. The kits employed were described. As you point out, I have no idea if MACS is the best way to go about this.
Further, please provide the detection limit of the assay used. Also, are the day-18 PFU values significantly above the detection limit? This influences interpretation of the findings.
Thank you for your comments.
Regarding your inquiry, the detection limit is 20 PFU/mL.
This experiment is to
- collect a tumor from each individual
- cut the tumor to about 1-2 g.
- homogenize 2.
- homogenize 2. and obtain supernatant
- measure PFU in the supernatant
The detection limit of PFU in the supernatant is determined by the process described in section 4. In addition, the limit of "PFU in a tumor cut into 1-2g" is calculated to be 10 PFU/tumor (1-2g). However, the detection limit in PFU/g cannot be given because the tumor volume differs from individual to individual.
The D18 results for the combination are shown in Fig. 4, as only one of the three samples had a result above the detection sensitivity.
By the way, time-points 13 and 18 are given relative to the tumor transplantation day. For interpretation, it is more relevant to present this relative to the day of virus injection.
Thank you for your comments. D13 is 2 days after the second virus administration and 2 days after the anti-PD-L1 antibody administration; D18 is 2 days after the third virus administration and 7 days after the anti-PD-L1 antibody administration. It is important to consider not only the timing from virus administration but also the number of days from antibody administration. It is difficult to complete all the information in Fig. I think the current expression is fine.
Discussion section:
From the observation that combination treatment reduced the amount of virus in tumors, while virus-specific CTL were detected, it is concluded on lines 214-216: “This suggests that the immune activation toward HSV and the tumor attenuated the antitumor effect of T-01 monotherapy.” The observation does not suggest this at all. In contrast, the experiment shown in Figure 2 showed that combination treatment was more effective than oHSV alone, on injected and non-injected tumors. By the way, the effect of PD-L1 Ab on HSV-specific CTL was not significant anyway (Figure 3b).
Thank you for your comment. I have corrected it.
The speculation on lines 236-239 about a presumed exhaustion of T cells upon “concomitant stimulation with T-01 and the associated tumor antigens” does not seem to be supported by the presented data. The experiment shown in Figure 5 was done in the absence of tumor cells. TAAs do thus not appear to play a role in the reduced recovery of PHA-stimulated lymphocytes upon T-01 infection.
Thank you for your comment. As indicated, the experiment in Fig. 5 does not reproduce in vivo conditions. However, we believe that the cytotoxicity of T-01 on activated lymphocytes by PHA is sufficient to estimate the in vivo situation.
Minor:
The description of the effects of E6/E7 on the immune system (lines 42-47) require supporting citations.
Thank you for your comment. Citation was added.
The statement that “Virus-related cancers are immunotherapy targets for neoantigens derived from cancer testis antigens […]” needs correction and explanation. (1) The neoantigens are the targets; (2) while this is evident for viral antigens, it is not immediately clear why virus-induced transformation is associated with cancer testis antigen expression.
Thank you for your comment. I have corrected it.
On line 67, the authors probably don’t mean to say that therapies are needed that “restore” the immunosuppressive TME. Presumably, they mean “overcome”, “counteract”, OSLT.
Thank you for your comment. I cannot understand “OSLT”.
Citation #10 does not sufficiently support the statement on lines 74-75 that oHSV shrank metastatic tumors by activating tumor immunity. In that study, oHSV armed with immune stimulatory transgenes were used. Best results were obtained when multiple immune stimulatory transgenes were combined. Effects of control empty oHSV on contralateral tumors were not significant.
Thank you for your comment. I have corrected it.
On lines 76-77, the suggestion is made that oHSV can turn cold tumors into hot tumors. The citation #11 to support this is however work with another type virus, not HSV. Please provide a relevant citation or delete the statement.
Thank you for your comment. I have deleted it.
On line 176, “T0-1” should be “T-01” and “of” should be “on”.
Thank you for your comment. I have corrected it.
In the legend of Figure 5, please clarify if “The experiments were performed four times” (line 188) means the data are the mean of 4 experiments; or are from one typical experiment of four done.
Thank you for your comment. This figure shows a summary of the results of four measurements.
In the discussion section 3, all references to figures are wrong. 2 should be 3; 3 should be 4; and 4 should be 5.
Thank you for your comment. I have corrected it.
The “2” in “W2” on line 299 should be written in superscript.
Thank you for your comment. I have corrected it.
Reference #32 on line 300 is superfluous; and probably not the appropriate citation for the applied formula, which was common practice long before the cited publication.
Lines 346-347: repeated sentence.
Thank you for your comment. I have corrected it.
Line 349: Tecon should read Tecan.
Thank you for your comment. I have corrected it.
Again, thank you for giving us the opportunity to strengthen our manuscript with your valuable comments and queries. We have worked hard to incorporate your feedback and hope that these revisions persuade you to accept our submission.
Sincerely,
Masahiro Kagabu M.D.
Department of Obstetrics and Gynecology
Iwate Medical University School of Medicine
2-1-1 Idaidori, Yahaba-chou, Shiwa, Iwate, 028-3695 Japan
E-mail: mkagabu@iwate-med.ac.jp
Tel: +81-19-611-8007; FAX: +81-19-907-67

Reviewer 2 Report
The manuscript is well written and organized, and comprehensive. The work is relevant to the journal's scope and has a clear scientific motivation. The references are up-to-date and figures are clear and appropriate. In general, the work is scientifically sound. Therefore, I recommend publication of this manuscript. Minor points:
* The aim of the work should more clarify in the introduction part.
* The resolution of figure 2 needs to be improved.
* Conclusion part should be rewritten to be supported by the experimental evidence supplied.
Reviewer 3 Report
General comments
In the manuscript by Kagabu M and co-workers, attempt was made to treat the murine model burden with HPV-related cervical cancer using third-generation oncolytic herpes simplex virus combined with immune checkpoint inhibitor, objectively showing that this kind of combination of therapy can markedly boost the number of tumor specific T cells in the tumor, with the limitation that T-01 had a cytotoxic effect on stimulated T cells, which leads to the suggestion that improved combination therapy can be achieved through optimizing the timing of administration.
On the whole, of particular translation is this report described. The writing was well-articulated, the experiment set was well controlled and the data shown were solid that were totally in support of the conclusions the authors came to. Given these, actually I have no more to say of. However, I would come off as not making any effort as a goalkeeper in this publication game, if I will not raise any question. So, despite the paper was sound and sounds good, there were still some questions I have labeled as minor that need to be addressed.
Minor questions I have
1. Could the authors provide some histopathological sections (including HE or IHC) reflecting the histological variation and contrast before and after combinatorial treatment?
2. How many mice were involved?
3. Using what method in measuring the volume of tumor lesions?
4. Here, the cervical cancer you referred to is squamous or adenoma? Please specify;
5. TC-1 cells were derived from murine lung epithelial nothing but co-transfected with HPV16 E6/7 and Ras, which seems to bear little on cervical cancer. Doesn’t it? At best, it is HPV related tumor not cervical cancer;
6. Further to clause 5, if so, the title had better be modified.

Round 2
Reviewer 1 Report
General:
While the main conclusion in the abstract and conclusion sections is corrected, the new wording seems not entirely correct. It is now stated on lines 26-27 and 361-363 that oncolytic HSV with ICIs “has a compensatory effect”. Compensation for what and on what? I think it is fair to say that the two agents partially counteract. A correct statement could e.g. be “These results suggest that T-01 and anti-PD-L1 Ab partially counteract and therefore concomitant administration should be considered with caution.” If the authors have support for activation of compensatory mechanisms, they should elaborate on these.
Specific:
Tumor growth inhibition experiment (Figures 1 and 2):
Figure 2 and corresponding text was mostly corrected appropriately. However, the new text on lines 104-105 seems incorrect. It reads “In addition, tumor growth was significantly inhibited by combination therapy compared to PD-L1 Ab therapy (Fig. 2 a and c).” Figure 2 however shows that the difference between these two treatment groups was not significant.
CTL analysis (Figure 3):
It is unfortunate that the authors cannot provide data, in particular tumor-specific CTL, on the control PBS group. While this omission should not preclude publication of the results, they should include text mentioning the missing control group and argue why this does not limit their conclusions.
The authors have “corrected” the discrepancy in the numbering of the days in the experiment, by changing “Eighteen days” to “Thirteen days” on line 146. This introduces a new discrepancy with the statement on line 127. They now state that they analyzed the samples 13 days after tumor implantation (line 146) and 13 days after T-01 inoculation (line 127). This is impossible, as T-01 was inoculated 7 days after tumor implantation (Figure 1).
Virus titer experiment (Figure 4):
The experiment is presented in figure 4 more clearly now.
Thank you for providing an explanation of the detection limit. Please also include this in the text of the manuscript, as the detection limit influences interpretation of the findings. Although an exact value cannot be given due to variable tumor fragment weights, based on your calculation it seems fair to say that the limit was 5-10 pfu/g. This implies that there is probably no significant difference between the BDL observations and the values detected on day 18.
As for the indication of the time-points in this experiment, it is fully acceptable to present days after the tumor transplantation in the figure. However, to help readers interpret the results it is relevant to also mention at least in the text how long this was after virus injection. And, if you consider this important, also to mention how long this was after antibody administration.
Discussion section:
The statement on lines 214-216 (now 210-212) is changed, but is still incorrect. It cannot be concluded that antitumor effect of T-01 in combination therapy with ICI compared to T-01 monotherapy is attenuated by immune activation. First and foremost, Figure 2 shows that combination treatment was significantly more effective than T-01 monotherapy. Second, while the reduced amount of virus in tumors indeed suggests immune activation against the virus and one may propose that this attenuates the antitumor effect, it is not shown and difficult to comprehend that an immune response against the tumor attenuates the antitumor effect. Please correct the statement.
The response to my remark on the speculation about a presumed exhaustion of T cells upon “concomitant stimulation with T-01 and the associated tumor antigens” is inadequate. I do not criticize the use of the in vitro culture, but disagree with the conclusion that this experiment, which was done in the absence of tumor cells, shows that TAAs play a role in the reduced recovery of PHA-stimulated lymphocytes upon T-01 infection. Simply because there were no TAAs in the experiment.
Minor:
The following minor points remain to be edited:
Line 60-61: cancers are not “immunotherapy targets for antigens”.
The revised legend to Figure 5 is not clear. What is “a summary of four experiments”? Does the figure not include all results from these experiments?
